# ONE THING TO FOOL THEM ALL: GENERATING INTERPRETABLE, UNIVERSAL, AND PHYSICALLY-REALIZABLE ADVERSARIAL FEATURES

## ABSTRACT

It is well understood that modern deep networks are vulnerable to adversarial attacks. However, conventional methods fail to produce adversarial perturbations that are intelligible to humans, and they pose limited threats in the physical world. To study feature-class associations in networks and better understand the real-world threats they face, we develop feature-level adversarial perturbations using deep image generators and a novel optimization objective. We term these *feature-fool* attacks. We show that they are versatile and use them to generate targeted feature-level attacks at the ImageNet scale that are simultaneously interpretable, universal to any source image, and physically-realizable. These attacks can also reveal spurious, semantically-describable feature/class associations that can be exploited by novel combinations of natural objects. We use them to guide the design of "copy/paste" adversaries in which one natural image is pasted into another to cause a targeted misclassification.

## 1 INTRODUCTION

State-of-the-art neural networks are vulnerable to adversarial inputs, which cause the network to fail yet only differ from benign inputs in subtle ways. Adversaries for visual classifiers conventionally take the form of a small-norm perturbation to a benign source image that causes misclassification (Szegedy et al., 2013; Goodfellow et al., 2014). These are effective, but to a human, these perturbations typically appear as random or mildly-textured noise. As such, analyzing these adversaries does not reveal information about the network relevant to how it will function–and how it may fail–when presented with human-interpretable features. Another limitation with conventional adversaries the limited extent to which they are physically-realizable. While they can retain some effectiveness when photographed in a controlled setting (Kurakin et al., 2016), they are less effective in uncontrolled settings (e.g Kong et al. (2020)) and cannot be created from natural objects.

Several works discussed in Section 2 have aimed to produce adversarial modifications that are universal to any source image, interpretable, *or* physically-realizable. But to the best of our knowledge, none exist for accomplishing all three at once. To better understand networks and what threats they face in the real world, we set out to create adversarial examples with all of these desiderata. Fig. 1 gives an example of one of our attacks in which a universal adversarial patch depicting a crane is physically placed near sunglasses to fool a network into classifying the image as a pufferfish.

Because pixel-space optimization produces non-interpretable perturbations, the ability to manipulate images at a higher level is needed. We take inspiration from recent advancements in generative modeling (e.g. Brock et al. (2018)) at the ImageNet (Russakovsky et al., 2015) scale. Instead of pixels, we perturb the latent representations inside of a deep generator to manipulate an image in feature-space. In doing so, we produce adversarial features which are inserted into source images either by directly modifying the latents used to generate them or by inserting a generated patch into a natural image. We combine this with a loss that uses an external discriminator and classifier to regularize the adversarial feature into appearing interpretable and not resembling the target class.

We use this strategy to produce what we term as *feature-fool* attacks in which a feature-level manipulation causes a misclassification yet appears intelligible to a human without resembling the target class. We show that our universal attacks are more interpretable and better disguised than analo-

**Sunglasses: 0.8101**
**Crane: 0.0**
**Puffer: 0.0**

**Sunglasses: 0.0**
**Crane: 0.9206**
**Puffer: 0.0**

**Sunglasses: 0.0107**
**Crane: 0.0**
**Puffer: 0.9783**

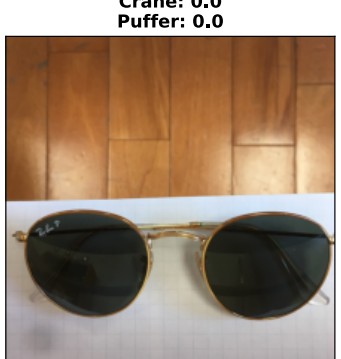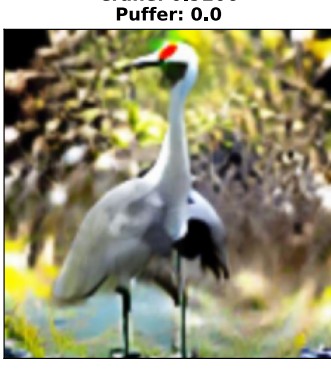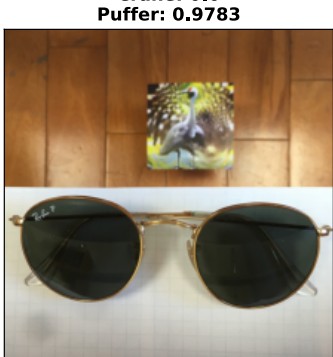

Figure 1: An example of an interpretable, universal, and physically realizable feature-fool attack against a ResNet50. The patch depicts a crane, however, when printed, physically inserted into a scene with sunglasses, and photographed, it causes a misclassification as a pufferfish. The patch was created by perturbing the latent of a generator to manipulate the image in a feature-space and training with a loss that jointly optimizes for fooling the classifier and resembling some non-target disguise class.

gous pixel-space attacks while being able to successfully transfer to the physical world. To further demonstrate their potential for interpretable and physically-realizable attacks, we also use these adversaries to guide the design of *copy/paste* attacks in which one natural image is pasted into another to induce an unrelated misclassification. Based on these findings, we emphasize the importance of cautious deployment for vision networks and their fortification against these types of feature-level adversarial attacks. The following sections contain related work, methods, experiments, and a discussion. For a jargon-free summary of our work for readers who are less familiar with research in deep learning, see Appendix A.7.

## 2 RELATED WORK

Conventional adversaries (Szegedy et al., 2013; Goodfellow et al., 2014) tend to be non-interpretable pixel-level perturbations and have limited ability to transfer to the physical world. Here, we contextualize our approach with other work and natural examples related to overcoming these challenges.

**Inspiration from Nature:** Mimicry is common in nature, and sometimes, rather than holistically imitating another species' appearance, a mimic will only exhibit particular features. For example, many animals use adversarial eyespots to stun or confuse predators (Stevens & Ruxton, 2014). Another example is the mimic octopus which imitates the patterning, but not the shape, of a banded sea snake. We show in Figure 2 using a photo of a mimic octopus from Norman et al. (2001) that a ResNet50 classifies it as a sea snake.

**Generative Modeling:** In contrast to pixel-space attacks, our method hinges on using a generator to manipulate images at a feature-level. One similar approach has been to train a generator or autoencoder to produce adversarial perturbations that are subsequently applied to natural inputs. This has been done by Hayes & Danezis (2018); Mopuri et al. (2018a;b); Poursaeed et al. (2018); Xiao et al. (2018); Hashemi et al. (2020); Wong & Kolter (2020) to synthesize attacks that are transferable, universal, or efficient to produce. Unlike these, however, we also explicitly focus on physical-realizability and human-interpretability. Additionally, rather than training an adversary generator, ours and other related works have skipped this step and simply trained adversarial latent perturbations to pretrained models. Liu et al. (2018) did this with a differentiable image renderer. Song et al. (2018) and Joshi et al. (2019) used deep generative networks, as well as Wang et al. (2020) who aimed to create more semantically-understandable attacks by training an autoencoder with a "disentangled" embedding space. However, these works focus on small classifiers trained on simple datasets (MNIST (LeCun et al., 2010), SVHN (Netzer et al., 2011), CelebA (Liu et al., 2015) and BDD (Yu et al., 2018)). In contrast, we work at the ImageNet (Russakovsky et al., 2015)

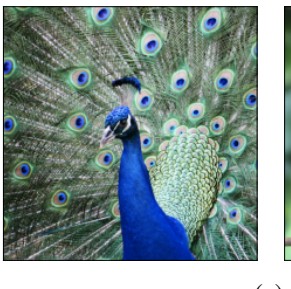 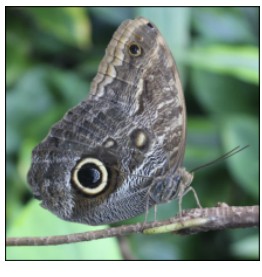 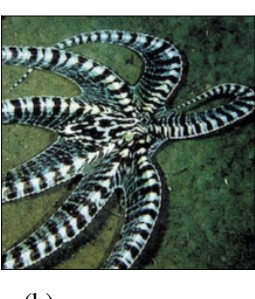

(a)                                                  (b)

Figure 2: Adversarial features in nature. (a) A peacock and butterfly with adversarial eyespots (unfortunately, peacock and ringlet butterfly are ImageNet classes, so one cannot meaningfully test how ImageNet networks might be misled by them). (b) A mimic octopus from Norman et al. (2001) is classified as a sea snake by a ResNet50.

scale. Hu et al. (2021) do this as well, but only using adversarial patches, while we do so with three types of attacks discussed in the following section. Finally, compared to all of the above works, ours is also unique in that we directly regularize adversaries for interpretability and disguise with our training objective rather than relying on perturbations in latent space alone.

**Physically-Realizable Attacks:** Our first contribution related to physical realizability is *interpretable* attacks that fool a classifier even when printed and photographed. This directly relates to the work of Kurakin et al. (2016) who found that conventional pixel-space adversaries could do this to a limited extent in controlled settings. More recently, Sharif et al. (2016); Brown et al. (2017); Eykholt et al. (2018); Athalye et al. (2018); Liu et al. (2019); Kong et al. (2020); Komkov & Petiushko (2021) used optimization under transformation to create adversarial clothing, stickers, patches, or objects for fooling vision systems. In contrast to each of these, we generate attacks that are not only physically-realizable but also inconspicuous in the sense that they are both interpretable and disguised. Our second contribution involving physically-realizable attacks is "copy-paste" attacks discussed next.

**Interpretable Adversaries:** In addition to fooling models, our adversaries provide a method for discovering semantically-describable feature/class associations learned by a network. This relates to work by Geirhos et al. (2018) and Leclerc et al. (2021) who debug networks using rendering and style transfer to make a zero-order search over features, transformations, and textural changes in images that cause misclassification. More similar to our work are Carter et al. (2019) and Mu & Andreas (2020) who develop interpretations of networks using feature visualization (Olah et al., 2017) and network dissection (Bau et al., 2017) respectively. Both find cases in which their interpretations suggest a "copy/paste" attack in which a natural image of one object is pasted inside another natural image to cause a misclassification as a third object. We add to this work with a new method to identify adversarial features for copy/paste attacks, and unlike either previous approach, ours naturally does so in a context-conditional fashion.

## 3 METHODS

### 3.1 THREAT MODEL

We adopt the "unrestricted" adversary paradigm of Song et al. (2018), which requires the network's classification of an adversarial example to differ from some oracle (e.g. a human). The adversary's goal is to produce a feature that will cause a targeted misclassification without resembling the target class. In particular, we focus on attacks that are universal across a distribution of source images and physically-realizable. We assume that the adversary has access to a differentiable image generator, a corresponding discriminator, an additional auxiliary classifier (optionally), and images from the target classifier's training distribution. We also require white-box access to the target classifier, though we present black-box attacks based on transfer from an ensemble to a held-out classifier in Appendix A.2. The adversary is limited in that it can only change a certain portion of either the latent or the image, depending on the type of attack we use.

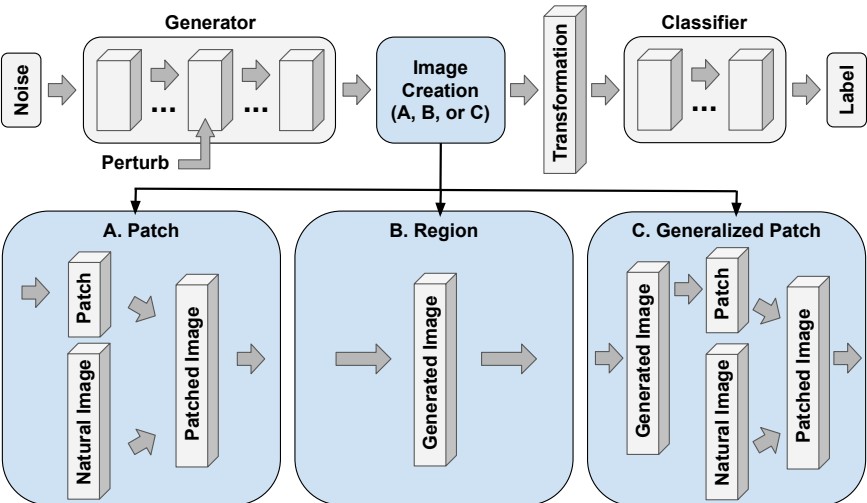

Figure 3: Our fully differentiable pipeline for creating *patch*, *region*, and *generalized patch* attacks.

## 3.2 TRAINING PROCESS

Our attacks involve manipulating the latent representation inside of a single generator layer to produce an adversarial feature. Fig. 3 outlines our overall approach. We produce three kinds of adversarial attacks, *patch*, *region*, and *generalized patch*:

**Patch:** We use the generator to produce a square patch that inserted in a natural image.

**Region:** We randomly select a square portion of the latent representation in a generator layer spanning the channel dimension but not the height or width dimensions and replace it with a learned insertion. This is analogous to a patch of the image in its pixel representation. The modified latent is then passed through the rest of the generator, producing the adversarial image.

**Generalized Patch:** This method produces a patch that can be of any shape, hence the name "generalized" patch. First, we generate an image in the same way that we do for region attacks. Second, we extract a generalized patch. We do this by (1) taking the absolute-valued pixel-level difference between the original and adversarial image, (2) applying a Gaussian filter for smoothing, and (3) creating a binary mask from the top decile of these pixel differences. We then apply this mask to the generated image to isolate the region of the image that the perturbation altered. We can then treat this as a patch and overlay it onto an image in any location.

**Objective:** For all attacks, we train a perturbation $\delta$ to the latent of the generator to minimize a loss that optimizes for both fooling the classifier and appearing as an interpretable, disguised feature:

$$\arg\min_{\delta} \mathbb{E}_{x\sim\mathcal{X},t\sim\mathcal{T},l\sim\mathcal{L}} \left[ L_{\text{x-ent}}(C(A(x,\delta,t,l)), y_{\text{targ}}) + L_{\text{reg}}(A(x,\delta,t,l)) \right]$$

with $\mathcal{X}$ a distribution over images or generator encodings of them (e.g. the validation set or generation distribution), $\mathcal{T}$ a distribution over transformations, $\mathcal{L}$ a distribution over insertion locations (which only applies for patch and generalized patch adversaries), $C$ is the target classifier, $A$ an image-generating function, $L_{\text{x-ent}}$ a targeted crossentropy loss for fooling the classifier, $y_{\text{targ}}$ the target class, and $L_{\text{reg}}$ a regularization loss for interpretability and disguise.

$L_{\text{reg}}$ contains several terms. Our goal is produce features that are interpretable and disguised to a human, but absent the ability to scalably or differentiably have a human in the loop, we instead use $L_{\text{reg}}$ as a proxy. All terms in $L_{\text{reg}}$ for each type of attack are listed in the following section, but most crucially, it includes ones calculated using a discriminator and an auxiliary classifier. For all three types of attack, we differentiably resize the patch or the extracted generalized patch and pass it through the discriminator and auxiliary classifier. We then add weighted terms to the regularization

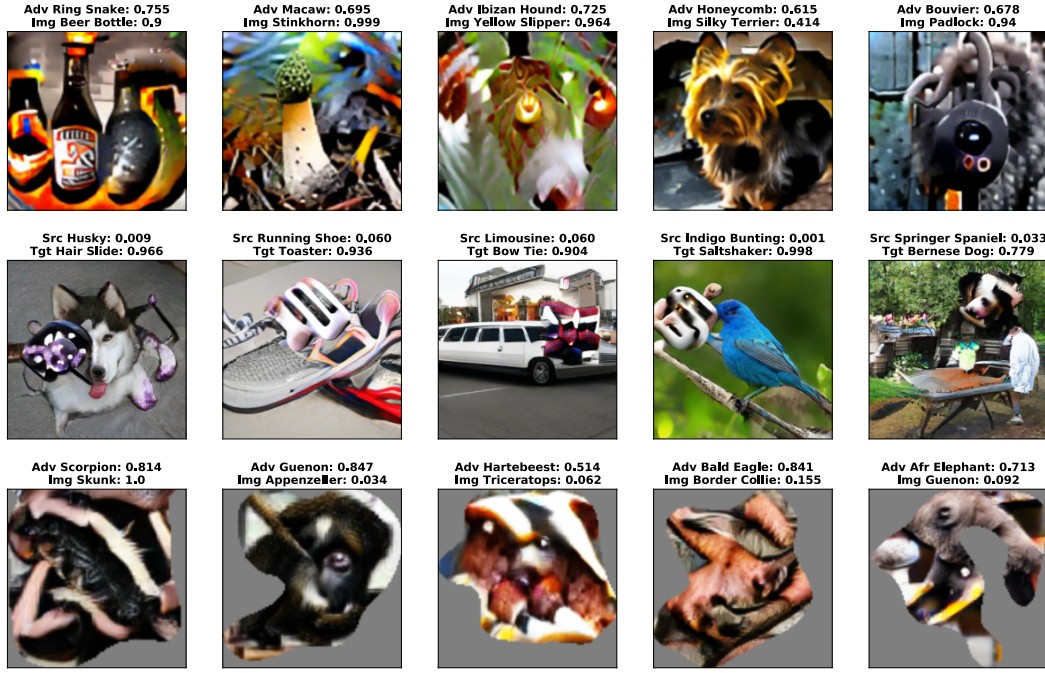

Figure 4: Examples of universal patch (top), region (middle), and generalized patch (bottom) feature-fool attacks. Each patch and generalized patch is labeled with its mean fooling confidence under insertion for random locations and source images (labeled 'Adv') and the confidence with which it is classified as the disguise class (labeled 'Img'). Numbers in each patch and generalized patch subfigure title come from different inputs, so they can add to be $> 1$. The region attacks are labeled with their confidence as their source class ('Src') and the target class ('Tgt').

loss based on (1) the discriminator's ($D$) logistic loss for classifying the input as fake, (2) the softmax entropy of a classifier's ($C'$) output, and (3) the negative of the classifier's crossentropy loss for classifying the input as the attack's target class. Thus, we have:

$$L_{\text{reg}}(a) = L_{\text{logistic}}[D(P(a))] + H[C'(P(a))] - L_{\text{x-ent}}[C'(P(a), y_{\text{targ}})] + \dots$$

Where $P(a)$ returns the extracted and resized patch from adversarial image $a$, and $\dots$ is a place-holder for additional terms explained in Section 4.1. These terms encourage the adversarial feature to (1) look real, and (2) look like some specific class, but (3) not the target class of the attack. The choice of what disguise class to use is left entirely to the algorithm.

## 4 EXPERIMENTS

### 4.1 ATTACK DETAILS

We use BigGAN generators off the shelf from Brock et al. (2018) using the implementation from Wolf (2018), and perturb the post-ReLU activations of the internal 'GenBlocks'. Notably, due to self attention inside of the BigGAN architecture, for region attacks, the change to the output image is not square even though the perturbation to the latent is. By default, we attacked a ResNet50 (He et al., 2016), and we restrict patch attacks to attacking 1/16 of the image and region and general-ized patch attacks to 1/8. We found that performing our crossentropy and entropy regularization on our patches using adversarially-trained auxiliary classifiers produced subjectively more inter-pretable results. Presumably, this relates to how adversarially-trained networks tend to learn more interpretable representations (Engstrom et al., 2019b; Salman et al., 2020) and better approximate the human visual system (Dapello et al., 2020). So for crossentropy and entropy regularization, we used a 2-network ensemble of an $\epsilon = 4$ $L_2$ and $\epsilon = 3$ $L_\infty$ robust ResNet50s from Engstrom et al. (2019a) for regularization. For discriminator regularization, we use the BigGAN class-conditional

discriminator with a uniform class vector input (as opposed to a one-hot vector). For patch adversaries, we train under colorjitter, Gaussian blur, Gaussian noise, random rotation, and random perspective transformations to simulate changes that a physically-realizable adversary would need to be robust to. For region and generalized patch ones, we only use Gaussian blurring and horizontal flipping. Also for region and generalized patch adversaries, we promote subtlety by penalizing the difference from the original image using the LPIPs perceptual distance (Zhang et al., 2018; So & Durnopianov, 2019). Finally, for all adversaries, we apply a penalty on the total variation of the patch or change induced on the image to discourage high-frequency, less-interpretable patterns. All experiments were implemented with PyTorch (Paszke et al., 2019).

Figure 4 shows examples of universal feature level patch, region, and generalized patch attacks. In particular, the patches on the top row are effective at resembling a disguise class to the network. (We also subjectively find that they resemble the disguise class to us as well.) However, when shrunk to the size of a patch and inserted into another image, the network sees them as the target class. This suggests size biases in how networks process features. And to the extent that humans also find these patches to resemble the target, this may suggest similar properties in the human visual system. However, it is key to recognize the framing effects when analyzing these images: recognizing target-class features given the target class versus given no information are different tasks (Hullman & Diakopoulos, 2011). Analyzing human perception of feature-level adversaries may be an interesting direction for future work.

## 4.2 INTERPRETABLE, UNIVERSAL, PHYSICALLY-REALIZABLE ATTACKS

To demonstrate that feature-fool adversaries are interpretable and versatile, we generate adversarial patches which appear as one object to a human, cause a targeted misclassification by the network as another, do so universally regardless of the source image, and are physically-realizable. We generated feature-fool patches using our approach and compared them to five alternatives. Four of which were ablation tests in which no generator was used in favor of optimization in pixel space and in which each of the three regularization terms discussed in Section 3.2 were omitted. In the final test, we omitted the generator and all three of the regularization terms, resulting in the same method as Brown et al. (2017). For each test, all else was kept identical including training under transformation and initializing the patch as an output from the generator. This initialization allowed for the pixel-space and Brown controls to be disguised and was the same as the approach for generating disguised pixel-space patch attacks in Brown et al. (2017).

**In Silico:** Before testing in the physical world, we did so in silico with 100 universal attacks of each type with random target classes. Fig. 5 plots the results. On the $x$ axis are target class

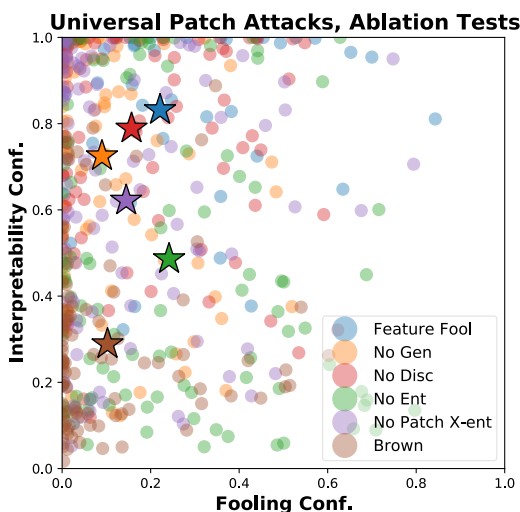

Figure 5: Feature-fool patch attacks produce more deceptive and interpretable targeted universal attacks than ablation tests. Fooling conf. gives target class confidence. Interpretability conf. shows the auxiliary network's label class confidence for the patch. Attacks further up and right are better. Centroids are shown as stars.

fooling confidences. On the $y$ axis are the labeling confidences from the auxiliary classifier which we use as a proxy for human evaluation. For all types of attacks, crafting small patches to be targeted, universal attacks has variable success for both fooling and interpretability. The centroids for all attacks are denoted with stars and suggest that our feature-fools are both better at fooling the classifier and appearing interpretable on average than all others with the exception of the adversaries trained without regularization on the entropy of the auxiliary classifier's output. However, the no-entropy regularization ablation adversaries had much lower interpretability confidences.

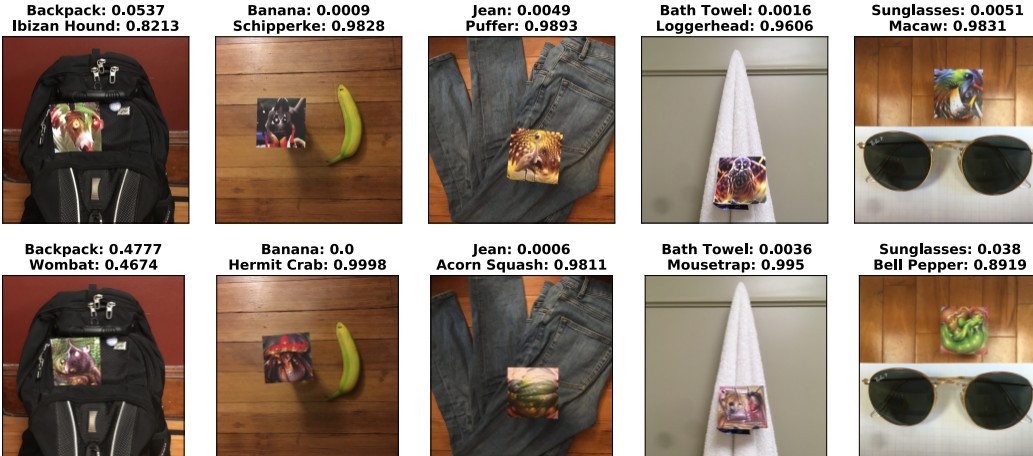

Figure 6: Successful examples of universal, physically-realizable feature-fool attacks (top) and pixel-space attacks (bottom). See Appendix A.6 for full-sized versions of the patches.

The fact that the feature-fool adversaries are more interpretable than controls without the entropy regularization (i.e. appear as some class by the auxiliary classifier) is unremarkable because they were specifically trained to do so. More notably though, the feature fool attacks fare significantly better than all but one of the others in mean fooling confidence–particularly the tests that did not use a generator. We also subjectively find the feature-fool patches to be more interpretable than the Brown controls. See Appendix A.6 for examples of feature-fool and Brown adversarial patches with high fooling confidences. Because they were initialized from generator outputs, some of the Brown patches have a veneer-like resemblance to non-target class features, but nonetheless, we find it clear from inspection that they contain higher-frequency patterns and are poorly disguised in comparison to our feature-fool attacks

**In the Physical World:** Next, we compared the physical realizability of our attacks with ones using the methodology of Brown we generated 100 additional adversarial patches for each, selected the 10 with the best mean fooling confidence, printed them, and photographed them next to 9 different ImageNet classes of common household items.[1] We confirmed that photographs of each object with no patch were correctly classified and analyzed the outputs of the classifier when the adversarial patches were added in the physical scene.

Figure 6 shows successful examples of these physically-realizable feature-fool and Brown patch attacks. Meanwhile, resizable and printable versions of all 10 feature-fool and Brown patches are in Appendix A.6. The mean and standard deviation of the fooling confidence for the feature-fool attacks in the physical world were 0.312 and 0.318 respectively ($n = 90$) while for the Brown attacks, they were 0.474 and 0.423 ($n = 90$). However, we do not attempt any hypothesis tests due to nonindependence between the results across classes due to the same set of patches being used for each class. These tests in the physical world show that the feature-fool attacks were often effective but that there is high variability in this effectiveness. The comparisons to Brown attacks provide some evidence that unlike our results in silico, the feature-fool attacks may be less reliably successful in the real world than the controls. Nonetheless the overall level of fooling success between both groups was comparable.

## 4.3 INTERPRETABILITY AND COPY/PASTE ATTACKS

Using adversarial examples to better interpret networks has been proposed by Dong et al. (2017) and Tomsett et al. (2018). Unlike conventional adversaries, feature-level adversaries reveal feature-class associations, which are potentially of greater practical interest. For experiments with interpreting a classifier, we use versions of our attacks in without the regularization terms described in Section 3.2.

---

[1]Backpack, banana, bath towel, lemon, jeans, spatula, sunglasses, toilet tissue, and toaster.

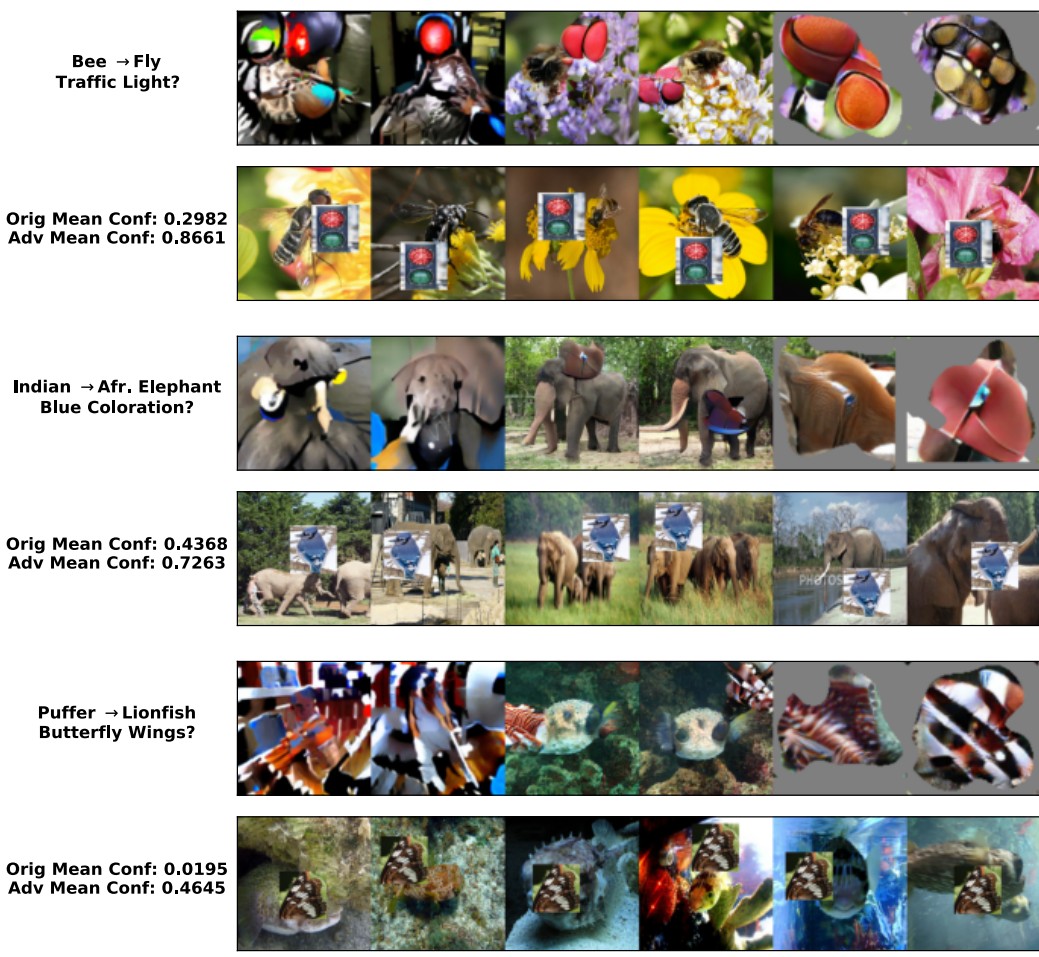

Figure 7: Patch, region, and generalized patch adversaries being used to guide three class-universal copy/paste adversarial attacks. Patch adversary example pairs are on the left, region adversaries in the middle, and generalized patch adversaries on the right of each odd row. Six successful attack examples are on each even row.

We find that inspecting the resulting adversarial features suggest both useful and harmful feature-class associations. In the Appendix, Fig. 11 provides a simple example of each.

**Copy/Paste Attacks:** A copy-paste attack is one in which a natural image is inserted into another in order to cause an unexpected misclassification. They are more restricted than the attacks in Section 4.2 because the features pasted into an image must be natural objects rather than ones whose synthesis can be controlled. As a result, they are of high interest for developing physically-realizable attacks because they suggest combinations of real objects that could yield unexpected classifications. They also have precedent in the real world. For example, feature insertions into pornographic images have been used to evade NSFW content detectors (Yuan et al., 2019).

To develop copy/paste attacks, we select a source and target class, develop class-universal adversarial features, and analyze them for common motifs that resemble natural objects. Then we paste images of these objects into natural images and pass them through the classifier. Two other works have previously developed copy/paste attacks, also via interpretability tools that discover feature-class associations: Carter et al. (2019) and Mu & Andreas (2020). However, compared to prior approaches, our technique may be uniquely equipped to produce germane fooling features. Rather than simply producing features associated with the target class, our adversaries generate fooling features *conditional* on the distribution, $\mathcal{X}$, over source images (i.e. the source class) with which the

adversaries are trained. This method allows any source/target classes to be selected, but we find the clearest success in generating copy/paste attacks when they are somewhat related (e.g. bee and fly).

Fig. 7 gives three illustrative examples. For each attack, we show two example images for each of the patch, region, and generalized patch adversaries. Below these are the copy/paste adversaries with average target class confidence before and after feature insertion for the 6 (out of 50) images for the source class in the ImageNet validation set for which the insertion resulted in the highest target confidence. Overall, little work has been done on copy/paste adversaries, and thus far, methods have always involved a human in the loop. This makes objective comparisons between methods difficult. However we provide examples of a feature-visualization based tool inspired by Carter et al. (2019) in Appendix A.4 to compare with ours.

## 5 DISCUSSION

By using a generative model to synthesize adversarial features, we contribute to a more pragmatic understanding of deep networks and their vulnerabilities. As an attack method, our approach is simple and versatile. Across experiments here and in the Appendix, we show that it can be used to produce targeted, interpretable, disguised, universal, physically-realizable, black-box, and copy/paste attacks at the ImageNet level. To the best of our knowledge, we are the first to introduce a method with *all* of these capabilities. As an interpretability method, this approach is also effective as a targeted means of searching for adversarially exploitable feature-class associations.

Conventional adversaries reveal intriguing properties of the learned representations in deep neural networks. However, as a means of attacking real systems, they pose limited threats outside of the digital domain (Kurakin et al., 2016). Given our feature-fool attacks, our copy/paste attacks, and related work, a focus on adversarial features and robust, physically-realizable attacks will be key to understanding practical threats. Importantly, even if a deep network is adversarially trained to be robust to one class of perturbations, this does not guarantee robustness to others that may be used to attack it in deployment. Consequently, we argue for focusing on pragmatic threats, training robust models (e.g. Engstrom et al. (2019a); Dapello et al. (2020)), and the use of caution with deep networks in the real world. As a promising sign, we show in Appendix A.5 that adversarial training is useful against feature-fool attacks.

A limitation is that when more constraints are applied to the adversarial generation process (e.g. universality + physical-realizability + disguise), attacks are generally less successful, and more screening is required to find good ones. They also take more time to generate which could be a bottleneck to using them for adversarial training. Further still, while we develop disguised adversarial features, we do not generally find them to be innocuous in that they often have somewhat unnatural forms typical of generated images. In this sense, our disguised attacks may nonetheless be detectable. Ultimately, this type of attack is limited by the efficiency and quality of the generator.

Future work should leverage new advances in generative modeling. One possibly useful technique could be to develop fooling features adversarially against a discriminator which is trained to recognize them from natural features. We also believe that studying human responses to feature-level adversaries and the links between interpretable representations, robustness, and similarity to the primate visual system (Dapello et al., 2020) are promising directions for better understanding both networks and biological brains. Ultimately, given that every single one of the 11 proposals for building safe, advanced AI outlined in Hubinger (2020) directly call for interpretability tools and/or adversarial robustness, we believe that continuing this type of work will be highly valuable for building AI systems that are more reliable and trustworthy.

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
