# OpenReview forum: "One Thing to Fool them All: Generating Interpretable, Universal, and Physically-Realizable Adversarial Features"
_ICLR.cc/2022/Conference — ICLR 2022 Submitted_

### Official Review · Reviewer_6FCU · 2021-10-31

**Correctness:** 3
**Technical Novelty And Significance:** 3
**Empirical Novelty And Significance:** 3
**Recommendation:** 5
**Confidence:** 4

**Main Review:**

The proposed method involves manipulating the latent representations inside of a single generator layer to produce an adversarial feature. Using these they propose three types of attacks: patch, region and generalized patch. The goal is targeted adversarial attack. There are multiple advantages of designing the attack like this. First, it allows a for humanly interpretable images. Next these can be applied to wide range of images. And importantly, unlike most adversarial work, these images are physically realizable (this is a big strength of this attack). The proposed targeted method also shows impressive performance in the experiments.

My main concern regarding the proposed method is how would the model perform under object detection and anomaly detection. For example, the attach show in attack 1 can be circumvented using object detection, which would crop two separate part of the image and hence mitigate the attack. The object detection approach might not work on all the showed examples, but then anomaly detection might hamper the progress. Given that the patch is visible easily, I assume this change will be recognizable to an anomaly detection method. That can also possibly render the attack useless. I would like to know the author's thoughts about these concerns.

**Summary Of The Paper:**

The authors propose an interpretable attack on DNNs that is also universal to source images and physically realizable as well. For this purpose they use generative model to synthesize adversarial features which are then added to the source image. The authors then show the efficacy of the method on experimental data.

**Summary Of The Review:**

The paper is well written and the ideas are novel. The main strength of the paper is the general applicability of the attack. However, as mentioned in the previous section, my main concern comes from how the method would perform in the presence of object detection and anomaly detection.

---

> ### Author Response · Authors · 2021-11-15
> **Response to 6FCU**
>
> Thank you for the constructive feedback.
>
> We think that segmentation/anomaly-detection-based defense methods would be useful for defending against our attacks. We know of some work that has investigated this type of defense method (e.g. Arnab et. al 2018, On the Robustness of Semantic Segmentation Models to Adversarial Attacks). And we ourselves test adversarial training as a defense in Appendix A.4.
>
> Ultimately, our focus is on the side of discovering vulnerabilities and interpreting the classifier, and no similar papers on attack methods of which we know on adversarial patches or features include experiments on segmentation/anomaly-detection-based defenses. We would leave this to future work.
>
> Finally, we would emphasize that the potential for these defense methods to avoid our attacks, should such theoretical defenses actually be built into models used in the real world,  does not do anything to diminish the usefulness of our methods for interpretability.

---

> > ### Comment · Reviewer_6FCU · 2021-11-21
> > **Response to Author Feedback**
> >
> > I appreciate the feedback provided by the authors. As acknowledged by the authors, my concerns still remain valid. Furthermore, the other reviewers also raised some valid concerns. Given these, I have decided not to update my score at the moment.

---

### Official Review · Reviewer_zy7u · 2021-11-02

**Correctness:** 2
**Technical Novelty And Significance:** 2
**Empirical Novelty And Significance:** 1
**Recommendation:** 3
**Confidence:** 5

**Main Review:**

[Strengths]

The paper is well motivated to generate adversarial patches that are human-interpretable and physically realizable.

[Weaknesses]

1. In Section 3.2, the description of the *region* adversarial attack is confusing and unclear. Does this attack mean that only a square region of the latent representation is perturbed and the other regions of the latent representation are not perturbed? Then how to determine which channels should be perturbed? Besides, it is also confusing that given the generated adversarial image, what is the corresponding original image? Can users generate adversarial images for a given original image, which may be a natural image?

2. The generation process of the binary mask in the *generalized patch* adversarial attack is ad-hoc. Why do the authors apply Gaussian smoothing before computing the binary mask? Why do the authors preserve the pixels with the top 10% pixel differences instead of the top 5% or top 20% pixel differences? The authors should explain the reasons for these settings.

3. In Figure 4, both the adversarial patches and adversarial images do not look like realistic/natural images. Such adversarial patches cannot meet the requirements in real applications. The authors mention that one goal is to make the generated patches look real, but in Figure 4, it is easy to tell that these patches are generated instead of real images.
Besides, many patches actually look like the target class, *e.g.* the second image in the first line looks like a macaw, and the third image looks like a hound. This conflicts with the disguise requirement.

4. The comparison between the proposed method and other methods is not convincing. First, baselines are not enough. The authors only compare the proposed method with one baseline method in Section 4.2. Second, the test images shown in Figure 6 are too simple to simulate the scenario in real applications, which weakens the validation of the effectiveness of the proposed method. In real applications, objects and scenes will be much more complex. It would be better if authors could test the effectiveness of their method on real-world images of intermediate-level complexity, instead of using extremely simple images.

5. The metric for interpretability in Section 4.2 is questionable. First, the prediction confidence cannot reflect the interpretability of images. For example, some fake OOD images may be classified with a high confidence. Thus, the prediction confidence measures the discriminative power of images, instead of interpretability. To this end, the loss of the discriminator may be more suitable, because the loss of the discriminator reflects whether the image looks real. Second, the comparison based on the prediction confidence is unfair. In the proposed method, the entropy of the classifier’s output is included in the loss function, which boosts the prediction confidence. However, the baseline method does not contain such a regularization term.

6. In Section 4.3, it is claimed that the adversarial patches help people to understand the feature-class associations learned by the DNN. However, the experiment just illustrates that the generated features can successfully attack the DNN, but cannot prove that such features are exactly what the DNN learns for the target class. The feature representation learned by a network needs to be more rigorously analyzed.

7. In odd rows of Figure 8, the generated patches are not realistic, as mentioned in the third weakness. And in even rows of Figure 8, the adversarial images generated using copy-paste attack are not realistic. For example, the traffic light can never appear on top of a bee. The authors are encouraged to add one more penalty term to make the final adversarial images (not the patch itself, but the natural image with the patch pasted) look realistic.

8. Some details of the proposed method are not clear.
- For three attacks in Section 3.2, I suggest authors use formulas or equations to describe the attacking method. Otherwise, it would result in ambiguities, especially for the generalized patch attack.
- In the first loss function in Section 3.2, inputs of the image-generating function include an input image $x$. I guess that for patch and generalized patch adversaries, $x$ refers to the natural image. However, for region attacks, what does $x$ refer to?
- In the second loss function in Section 3.2, there is an ellipsis in the end, which is confusing. What else does the loss include?
- In Section 4.1, the use of a 2-network ensemble as the auxiliary classifier is ad-hoc. The authors are encouraged to clarify the reason for using an ensemble instead of a single network.
- In Section 4.1, the authors mention two additional penalty terms, but do not give their explicit formulae, which hurts the reproducibility of the paper. Specifically, the authors mention that “for region and generalized patch adversaries, we promote subtlety by penalizing the difference from the original image using the LPIPs perceptual distance” and “for all adversaries, we apply a penalty on the total variation of the patch or change induced on the image.” However, there are no formulae given for these two penalty terms.


**Summary Of The Paper:**

This paper focuses on the problem of generating human-interpretable and physically realizable adversarial patches. The authors propose three kinds of adversarial attacks based on feature-level perturbations of the latent representation of generative models. The authors also try to reveal feature-class associations using the proposed attacks. The paper is well-motivated, but many details are not introduced clearly, and experimental results are not convincing.

**Summary Of The Review:**

The paper is well-motivated, but many details are not introduced clearly, and experimental results are not convincing. Some claims are not supported by their experimental results. Thus, I think the paper should be rejected.

---

> ### Author Response · Authors · 2021-11-15
> **Response to zy7u**
>
> Thank you for the constructive feedback.
>
> 1. Yes. We selected the square regions randomly, and do not perturb anything else. We will revise our description with attention to making this more clear. For attacking a natural image, we would need to use a class-universal patch or generalized patch attack rather than a region attack.
>
> 2. We chose gaussian smoothing and the percentiles we did because it worked well for patch processing in practice. There is no deeper justification for it in the same way that there is no deep justification for why we use the learning rates we do instead of ones that are a few times higher or lower. These are fairly small details that aren’t key to the overall procedure, and we will consider moving them to the appendix.
>
> 3. We find that the realisticness of our adversarial patches is comparable to that of BigGAN outputs in general, suggesting that this concern is more one about generators than our approach. Generating reliably photorealistic images from a distribution similar to ImageNet’s is still an unsolved problem. However is not our aim l to produce photorealistic adversarial features, and there exist realistic situations in which this would not be required, such as graphics on a t-shirt. Our goal was to produce features that can, at a high level, be described, understood, and studied by humans. As for potential semblance of our adversarial patches to the target class, the last paragraph of 4.1 was somewhat meant to address this. These patches may resemble the target class, but (a) they are certainly better disguised than controls, and (b) we think it is key to emphasize that recognizing the target class when it is known is a very different task than recognizing it prima facie.
>
> 4. First see our response to 1mBx. We are conducting some new ablation tests. But aside from these, are there additional baselines other than the Brown et al that we should used in 4.2? Concerning figure 6, we our goal is a reliable comparison to a baseline rather than testing the limits of our adversaries’ ability to fool classifiers when photographed in the real world. Ultimately, robust physical realizability remains challenging, and we know of no other works that reliably produce adversarial objects that fool a classifier under arbitrary viewpoints without sampling and cherry picking. Most importantly,t our goal is not to produce adversarial attacks that are uniquely good at being physically realizable, but rather attacks that have a unique combination of properties including physical realizability.
>
> 5. Prediction confidence does not imply interpretability, but we find that the discriminative power of the images reliably associates with the auxiliary classifier’s ability to see recognizable features in them in our case. Also note that this classifier is adversarially trained which makes it less liable to produce confident labels from OOD inputs. You mention that we should use the loss of the discriminator, which we already do as described in the last paragraph of 3.2.
> It is true that we don’t include a control that uses regularization but not the optimization in latent space. This is fair in the sense that Brown et. al, 2017 did not do this either. But regardless, we are working on ablation tests to investigate this as mentioned in our response to 1mBx.
>
> 6. We think that saying “the experiment just illustrates that the generated features can successfully attack the DNN, but cannot prove that such features are exactly what the DNN learns for the target class” may be a distinction without a practical difference. Our viewpoint is essentially that a successful attack is a successful attack and therefore reveals something about the network’s vulnerabilities.
>
> 7. Regardless of whether the adversaries in the odd rows of figure 8 are “realistic” by any particular standard they still led us to successful copy paste attacks--we think the proof is in the pudding. Regardless of whether a traffic light could appear next to a bee, this *type* of attack with novel combinations of real objects, is of clear interest to interpretability and real world robustness (e.g. Carter et. al, 2019, and Mu et al, 2020).
>
> 8. We think that these five bullet points are good suggestions, especially {2,3,5}, and will rework the methods section for clarity.

---

> > ### Author Response · Authors · 2021-11-16
> > **Addition to point 5.**
> >
> > In response to
> >
> > "Second, the comparison based on the prediction confidence is unfair. In the proposed method, the entropy of the classifier’s output is included in the loss function, which boosts the prediction confidence. However, the baseline method does not contain such a regularization term."
> >
> > We know this and say so. On the end of page 6 and beginning of page 7, we write:
> >
> > "That the feature-fool adversaries are more interpretable (i.e. appear as some class by the auxiliary classifier) is unremarkable because they were specifically trained to do so. More notably though, this accompanied an increase in mean fooling confidence. We also subjectively find the feature fool patches to be more interpretable than the pixel-space ones. See Appendix A.5 for examples of feature-fool and pixel-space adversarial patches with high fooling confidences."

---

### Official Review · Reviewer_1mBx · 2021-11-02

**Correctness:** 3
**Technical Novelty And Significance:** 4
**Empirical Novelty And Significance:** 2
**Recommendation:** 6
**Confidence:** 4

**Main Review:**

Strengths:
- The result that adversarial patches can be created that look like a different class to the target (and are classified as such) is a novel, interesting and important result.
- The paper includes plenty of visual examples to demonstrate that the attacks are interpretable in the way they claim

Weaknesses:
- The paper contains very few quantitative experiments. The only comparison to existing methods is to compare the average fooling confidence and interpretability confidence of their adversarial patches against Brown et al patches. No quantitative experiments were provided for "region" and "generalised patch" attacks. To improve this more quantitative experiments should be included in the paper. Useful experiments would be ablation studies testing different combinations of pixel vs latent, discriminator vs no discriminator and entropy loss vs no entropy loss. Even better would be to include one or two more comparisons from some of the literature cited in Section 2.

**Summary Of The Paper:**

This paper introduces an adversarial patch attack in which the patches are classified with high confidence as a different class to the target when classified alone. They are also visually interpretable by humans as this other class, in contrast to conventional adversarial patches, which often can be easily interpreted as the class they are targeting. The core technical contribution of this paper is the introduction of an explicit regularisation to cause the patch to be classified differently to the target class.

**Summary Of The Review:**

The core idea of the paper is interesting and novel. I am concerned by the limited experiments, but this does not cause me to doubt the author's main claims. I lean very slightly towards accept but I think the paper might be better suited to a workshop.

---

> ### Author Response · Authors · 2021-11-15
> **Response to 1mBx**
>
> Thank you for the constructive feedback.
>
> Re: “The paper contains very few quantitative experiments.”
>
> We agree that more quantitative experiments would strengthen the paper, and we are currently working on ablation tests with the goal of sharing results here before the discussion deadline. However, we also argue that much excellent work involving techniques for developing human interpretable understandings of networks focuses on qualitative, visual results (e.g. Olah et. al 2017 or Carter et. al 2019).

---

> ### Comment · Reviewer_1mBx · 2021-11-29
> **Keeping my score the same**
>
> Having read the other reviews, the response from the authors and the updated paper I do not feel as if my concerns were sufficiently addressed to warrant increasing my score.

---

### Official Review · Reviewer_k6Su · 2021-11-03

**Correctness:** 3
**Technical Novelty And Significance:** 2
**Empirical Novelty And Significance:** 2
**Recommendation:** 5
**Confidence:** 3

**Main Review:**

Strengths:
- Their method can synthesize adversarial examples that are universal to any source image, uses interpretable features, and physically realizable, which is novel as far as I know.

Weaknesses:
- The motivation of this study is unclear. They mention two points: "better understanding networks" and "what threats they face in the real world", but here are my responses.
	- "Better understanding networks":
		- They motivate their method "as a means of better understanding networks" but the relevant section (I assume 4.3) only mentions that "feature-level adversaries reveal feature-class associations, which are potentially of greater practical interest."  Existing feature visualization techniques can reveal feature-class associations. What new information do we gain by the proposed method in terms of understanding networks? (For instance, I believe we can use existing feature visualization techniques to produce something similar to examples in Figure 7. )
	- "what threats they face in the real world"
		- Why makes it significant to being able to produce adversarial perturbations that are intelligible to humans? They motivate this by saying it poses more threat in the physical world. In my view, the whole reason why adversarial examples are considered to be a threat is that these perturbations are almost imperceptible to humans, which makes it hard to detect even by humans, let alone computer vision systems. If perturbations are intelligible to humans, at least humans can detect these adversaries, which makes  less threatening in some sense, so I fail to understand their argument that intelligible perturbations pose more threat. Can you elaborate on this point?
		- They claim that their findings "emphasize the importance of cautious deployment for vision networks" but that is something the community is already aware of, so it's not a strong argument to make to convey the significance of their work.


- Nitpick: In the introduction, the authors note that "they pose limited threats in the physical world" and "they are generally ineffective in less controlled ones such as those experienced by autonomous vehicles", but recently [1] shows that it's easy to attack self-driving cars with adversarial examples in the real world.

[1] : Sato et al "Dirty Road Can Attack: Security of Deep Learning based Automated Lane Centering under Physical-World Attack"

**Summary Of The Paper:**

They propose feature-fool attack, a method to synthesize adversarial examples that are can be placed onto any source image, uses interpretable features, and physically-realizable.
They achieve this by perturbing latent representations of large scale generative models.
In doing so, they regularize adversarial features so that they appear interpretable but not similar to the attack's target class.
They apply their method to copy/paste attacks.

**Summary Of The Review:**

I think this paper has some interesting points, but it could be improved by making the motivation clear.

---

> ### Author Response · Authors · 2021-11-15
> **Response to k6Su**
>
> Thank you for the constructive feedback.
>
> 1. Re: "Existing feature visualization techniques can reveal feature-class associations. What new information do we gain by the proposed method in terms of understanding networks?"
>
> The first advantage that our method has over features visualization is that our optimization procedure can disguise the adversarial feature. Our results in 4.2 suggest that this makes them better for designing *hidden* patch attacks than previous methods.
>
> The second advantage is that we produce features in 4.3 that cause misclassification as the target class *conditional* on the distribution of source images under which we train the adversary. This allows for visualization of a more specific set of features: those that convert one class into another, rather than just those features that exemplify a particular class. We discuss this more in paragraph 2 on page 8. And in Appendix A.3, we directly compare our method to using feature visualizations. In the case of Indian and African elephants, our method suggests adversaries that feature visualization seems to offer no hints of.
>
> As for figure 7, we agree that it does not provide information that feature visualization couldn’t. We only present it as an example. We debated about including it for this reason. Please let us know if you think it would be better to remove it from the main paper.
>
> 2. Re: "Why makes it significant to being able to produce adversarial perturbations that are intelligible to humans?"
>
> One way in which an attack could be insidious is by being imperceptible to humans. Another is by being seen by humans for something that it is not. For example, the patch of a crane making an image look to a classifier as a pufferfish suggests vulnerabilities involving robust adversaries that hide in plain sight. Essentially, our technique provides a novel type of threat.
>
> Moreover, our experiments with copy/paste attacks show that our technique can guide the creation of adversarial attacks made just by juxtaposing combinations of objects, which is only possible if the human understands what adversarial feature the classifier was responding to.
>
> 3. Re: “They claim that their findings "emphasize the importance of cautious deployment for vision networks" but that is something the community is already aware of.”
>
> We will reword this sentence to better reflect this, but we don’t think this should be seen as a significant flaw in the paper.
>
> 4. Re: Nitpick
>
> Like a few of the other papers mentioned in related work (e.g. Athalye et. al 2018 or Brown et. al 2017), Sato et. al 2021 don’t take the naive approach of Kurakin et. al 2016 and instead optimize under transformation. As a result, this is not the kind of approach we mean when we refer to “conventional” adversaries. We will rework our discussion of this to avoid any confusion or inaccuracy. Ultimately, we don’t claim that our attacks are uniquely good at being physically realizable, but rather that they have a unique combination of capabilities including physical realizability.

---

> > ### Comment · Reviewer_k6Su · 2021-11-30
> > **Response**
> >
> > Thank you for your response. The motivations are now clearer. I've read other reviewers' responses and I agree with some of their concerns. Based on these, I keep my score.

---

### Author Response · Authors · 2021-11-17
**Global response: rebuttal revision**

1. In response to comments from zy7u, we have clarified descriptions of the regularization methods in section 3.2

2. Based in part on the “nitpick” mentioned by k6Su, we have reworked the discussion about previous works and physically realizable attacks to better reflect that this is a paper about attacks that are not unique by being physically realizable but rather by having a unique combination of properties including physical realizability. We have also modified the introduction and related works sections to clarify the fact that our work has implications for physical realizability both via patch/region/gen-patch attacks and via copy/paste attacks.

3. What used to be Figure 7 (bikinis and barbershops) has been moved to the appendix, in part due to feedback from k6Su.

4. In response to feedback from 1mBx and zy7u, we replaced figure 5 (the scatterplot) with a new one that includes data from more methods including ablation tests. This adds to the quantitative results that we feature and provides an empirical justification for using a generator plus each of the three regularization terms covered in Section 3.2. As before, it still compares our patches to ones created using the methodology of Brown et. al, 2017. The discussion in Section 4.2 has also been changed accordingly.

5. The supplemental code has undergone additional commenting for clarity and ease of use but hasn’t undergone any changes to code.

6. Finally, in response to comments from zy7u involving an apparent semblance to the target class in the adversarial patches from figure 4, we have added a PDF to the supplemental material offering a friendly challenge to zy7u to recognize the target class in the patches **before** knowing it. Instructions are in the document. Please let us know your thoughts in a reply.

Once again thank you for your feedback. We feel that it has been helpful. Regardless of whether this paper is accepted, we feel that your feedback has led to improvements.

---

### Decision · Program_Chairs · 2022-01-20

**Decision:**

Reject

**Comment:**

In this manuacript, the authors develop feature-fool attacks with feature-level adversarial perturbations using deep image generators and a novel optimization objective. They further show that the feature-fool attacks are versatile and can generate targeted feature-level attacks at the ImageNet scale that are simultaneously interpretable, universal to any source image, and physically-realizable.
The reviewers agree that the paper is well-motivated and the authors have addressed some concerns.
However, the reviewers still do not satisfy with some concerns so as to keep the initial scores.
In comparison with the manuscripts I'm handling, I have to recommend to reject it!